# Ammonium Polyphosphate with High Specific Surface Area by Assembling Zeolite Imidazole Framework in EVA Resin: Significant Mechanical Properties, Migration Resistance, and Flame Retardancy

**DOI:** 10.3390/polym12030534

**Published:** 2020-03-02

**Authors:** Jingyu Wang, Hui Shi, Pinlie Zhu, Yuanjie Wei, Jianwei Hao

**Affiliations:** 1School of Materials Science and Engineering, Beijing Institute of Technology, Beijing 100081, China; quillwang@163.com (J.W.); shihuia@mail.ustc.edu.cn (H.S.); 15636125761@163.com (P.Z.); weiyuanjia21@163.com (Y.W.); 2National Engineering Technology Research Center of Flame Retardant Materials, Beijing 100048, China

**Keywords:** zeolite imidazole framework, APP, interface assembly, flame retardant

## Abstract

A zeolite imidazole framework (ZIF-67) was assembled onto the surface of ammonium polyphosphate (APP) for preparing a series multifunctional flame-retardant APP-ZIFs. The assembly mechanism, chemical structure, chemical compositions, morphology, and specific surface area of APP-ZIFs were characterized. The typical APPZ1 and APPZ4 were selected as intumescent flame retardants with dipentaerythritol (DPER) because of their superior unit catalytic efficiency of cobalt by thermogravimetric analysis. APPZ1 and APPZ4 possessed 6.8 and 92.1 times the specific surface area of untreated APP, which could significantly enhance the interfacial interaction, mechanical properties, and migration resistance when using in ethylene-vinyl acetate (EVA). With 25% loading, 25% APPZ4/DPER achieved a limiting oxygen index value of 29.4% and a UL 94 V-0 rating, whereas 25% APP/DPER achieved a limiting oxygen index value of only 26.2% and a V-2 rating, respectively. The peak of the heat release rate, smoke production rate, and CO production rate respectively decreased by 34.7%, 39.0%, and 40.1%, while the char residue increased by 91.7%. These significant improvements were attributed to the catalytic graphitization by nano cobalt phosphate and the formation of a more protective char barrier comprised of graphite-like carbon.

## 1. Introduction

Ethylene-vinyl acetate (EVA) resin, which has excellent mechanical properties, an ease of processing, and chemical resistance, is widely applied in hot melt adhesives, biomedical devices, and insulating cable materials fields [1,2,3]. Nevertheless, EVA resin is extremely flammable and releases a high emission of heat and smoke, making flame-retardant modification an urgent concern [4,5]. For environmental concerns, intumescent flame retardants (IFRs) with low-toxicity, low smoke, and high efficiency are acceptable halogen-free flame retardants for EVA composites [6,7]. Among the components of the hybrid IFR system, phosphor-containing flame retardants such as ammonium polyphosphate (APP) and its derivatives are the most widely used acid source [8,9,10]. However, the low compatibility between APP and polymer materials resulted in the emigration of APP and limited its flame retardancy especially in some extreme environments [11,12]. Therefore, it is necessary to develop a highly efficient IFR system, contributing to the better compatibility and flame-retardant efficiency when used in EVA composites. 

Generally, incorporating high-efficient synergists into an IFR system can improve its flame-retardant efficiency, thereby decreasing the required loading of IFR to meet forced mechanical and flame-retardant demand [13,14]. Among these synergists used in an IFR system, transition metal oxides, hydroxides, and salts frequently exert high efficiency due to the catalytic carbonization effect [15,16]. The valence electrons of transition metals are generally distributed in the d orbital in the secondary outer layer. The outermost shell has only one or two electrons, which are easy to be lost and can accept lone electron pairs of ligands, thereby forming complexes [17]. These complexes containing transition metals are often unstable, which can act as intermediates for coordination catalysis during the reaction process, therefore enhancing their catalytic efficiency. In the previous research studies, transition metals synergists are often combined with an IFR system to improve the heat resistance of residual carbon structures by catalyzing the formation of graphite-like aromatic carbon in the condensed phase [18,19]. In addition, transition metals such as Cu, Co, and Ni are easy to form ligand complexes and chelates, so they are also widely used in the surface assembly modification of traditional flame retardants to incorporate with multifunctional characteristics [20].

In recent decade, metal organic frameworks (MOFs) comprised of transition metals [21,22] have attracted extensive attention in the fields of drug transmission, energy materials, and sensors due to their large specific surface area, polymeric porous structure, and adjustable ligands [23,24,25]. Bivalent transition metals commonly used in MOFs, including zinc [26], cobalt [27], copper [28], and nickel [29], have been identified as synergists to promote the catalytic carbonization of flame-retardant systems. Among these MOFs, zeolite imidazole frameworks (ZIFs) with unique divalent transition metals have become the focus of flame-retardant polymer applications. ZIFs were directly used as flame retardants [30,31,32] or assembled with other flame retardants to prepare new multifunctional additives [32,33,34], which can significantly increase the flame retardancy of polystyrene, epoxy resin, and polylactic acid at even low loading amounts. Furthermore, ZIF-8 and ZIF-67 are very sensitive to acidic conditions, and they totally degrade due to the protonation of the imidazole ligand by H^+^ in acidic solutions [33,34,35,36], which can be used as templates for the engineering of nanoporous materials [37]. Inspired by this, we intended to assemble ZIF-67 onto APP in water solution, and we obtain various APP by adjusting the hydrolysis reaction and assembled content of ZIF-67. NH4^+^ causes the protonation of the imidazole ligand from APP in weakly acidic solutions, etching the surface of APP and increasing its specific surface area. Meanwhile, the residual Co^2+^ and undecomposed ZIF-67 can improve the flame-retardant efficiency of APP. In the light of above assumptions, we develop a feasible strategy for rapidly modifying APP with a high specific surface area by assembling a zeolite imidazole framework. When using in an EVA matrix, the significant mechanical properties, migration resistance, and flame retardancy of the EVA composites were obtained. More importantly, the interface assembly of ZIF-67, which increases the specific surface area and the interaction with the polymer matrix, opens a new avenue for preparing APP with both high flame-retardant efficiency and migration resistance.

## 2. Experimental

### 2.1. Materials 

Co(NO_3_)_2_·6H_2_O and 2-methylimidazole (2-MIZ) were purchased from Aladdin Industrial Corp. Ammonium polyphosphate (APP, *n* > 1000) and dipentaerythritol (DPER) were obtained from Shanghai Macklin Biochemical Co., Ltd., Shanghai, China. Ethanol (98%) was purchased from Beijing Tongguang Fine chemicals Co., Beijing, China. Poly (ethylene vinyl acetate) copolymers (EVA18) were supplied as pellets by DuPont Co. (Elvax, 18 wt % of vinyl acetate, Wilmington, NC, USA). The maleic anhydride grafted EVA (gEVA, 30% of maleic anhydride) was purchased from Nanjing Huadu Science and Technology Industrial Co. Ltd., Nanjing, China.

### 2.2. Preparation of ZIF-67 and APP-ZIFs

ZIF-67 was prepared in the following steps: 2 mmol of Co(NO_3_)_2_·6H_2_O and 8 mmol of 2-MIZ were dissolved in 50 mL of ethanol solution, respectively, and then mixed. After 10 min magnetic stirring, the mixture was left to stand 24 h. Its precipitate was isolated by centrifuge, washed with deionized water and ethanol, and then dried for 4 h at 80 °C. In the typical preparation process of APP-ZIFs (Appendix A), APPZ1 was prepared in the following steps: 0.04 mol (250 mL) Co(NO_3_)_2_·6H_2_O was dissolved in ethanol solution. Then, 0.16 mmol (250 mL) 2-MIZ solution was added and stirred for 10 min. After this process, APP slurry (100 g APP in 200 mL deionized water) was added into the ZIF-67 precursor solution and stirred for another 2 h. During this procedure, ammonia-smelling gas was released from the solution due to the reaction between 2-MIZ and APP. Products were obtained by filtration and purification with water and ethanol three times, respectively. Finally, the APPZ1 was obtained after drying for 4 h at the temperature of 80 °C. Similarly, APPZ2, APPZ3, and APPZ4 were obtained using double, triple, and quadruple loading of cobalt nitrate and 2-methylimidazole solution per 100 g APP, respectively. 

### 2.3. Preparation of Flame-Retardant EVA Composites

Among the APP-ZIFs, the APPZ1 and APPZ4 were selected as flame retardants according to the better unit catalytic efficiency of cobalt by the thermogravimetric analysis, and the intumescent flame-retardant system used into EVA composites were obtained by blending APP, APPZ1, and APPZ4 with DPER in a ratio of 3:1. The flame-retardant EVA composites were prepared by twin-roll mixer at 120 °C for 10 min. The sheets with different thickness were prepared by compression molding under a pressure of 10 MPa at 120 °C for 10 min. The samples were labeled according to the content of flame retardants listed in Appendix A, while the control sample named as EVA was prepared using EVA18 containing 5% gEVA in the same manner.

### 2.4. Characterization

The surface elemental compositions of the flame retardants (FRs) and char residues were analyzed by X-ray photoelectron spectroscopy (XPS) on a PHI Quantera-II SXM (Ulvac-PHI, Inc., Chigasaki, Japan) using Al ka radiation and an X-ray power of 2.5 kW under a vacuum of 2.6 × 10^−7^ Pa. The cobalt contents of FRs were detected by the inductively coupled plasma mass spectrometry (ICP-MS) (iCAP Q, Thermo, Waltham, MA, USA). The crystal structures of the ZIF-67, APP, APPZ1, and APPZ4 were analyzed by X-ray diffraction (XRD), which was performed with a rotating anode X-ray diffractometer (Japan Rigaku D/Max-Ra, Tokyo, Japan) equipped with a Cu Kα (λ = 0.1542 nm) radiation at 2θ values ranging from 10° to 40°. Fourier transform infrared (FTIR) spectroscopy was performed with an ATR IR spectrometer (Nicolet 6700, Waltham, MA, USA). 

The micromorphology images of FRs and the brittle fracture of EVA composites and residues after a cone calorimeter were obtained using an FEI Quanta 250 FEG field-emission scanning electron microscope (SEM) at high vacuum conditions with a voltage of 15 kV. The element contents of char residues were investigated via an AMETEK Quanta 250 FEG/EDS energy-dispersive spectrometer (EDS, FEI, Inc., Hillsboro, OR, USA). Transmission electron microscopy (TEM) images of FR particles were obtained on FEI Tecnai G2 F30 (FEI, Inc., Hillsboro, OR, USA) with an acceleration voltage of 300 kV. 

The N_2_-sorption isotherms for APP-ZIFs particles were performed at 77 K, using a Micromeritics ASAP 2020 HD88 system utilizing Barrett–Emmett–Teller (BET) calculations for surface area.

Thermogravimetric analysis (TGA) under both the nitrogen and air atmosphere of FRs and flame-retardant EVA composites were performed with a thermal analyzer (Netzsch 209 F1, Netzsch, Selb, Germany) at a heating rate of 10 °C/min from 50 to 700 °C. The real-time FTIR spectra of evolved pyrolysis gases were collected via an FTIR spectrometer produced by PerkinElmer. The FTIR specimen cell and transfer line were both kept at 280 °C. 

The dynamic rheological properties of all the samples were tested using a rotational rheometer (ARES Rheometer, TA Inc., New Castle, DE, USA) at 150 °C with a pair of parallel plates (20 mm in diameter with a gap of 1.0 mm). The frequency range was 0.1–100 rad/s, and the maximum strain was fixed at 5%. The tensile properties were conducted at room temperature on a universal testing machine (CMT-410 4, MTS Systems Co., Ltd., Shanghai, China) with a cross-head speed of 50 mm/min following the ASTM-D412 standard. 

To determine the water resistance of the EVA composites, the specimens were put into distilled water at 70 °C and were kept at this temperature for 3 to 15 days, and the water was replaced every 24 h. The treated specimens were subsequently taken out and dried in a vacuum oven at 80 °C for 72 h, and the mass loss percentages were measured.

The limiting oxygen index (LOI) was measured by a FTT (Fire Testing Technology, London, UK) Dynisco LOI instrument with specimen dimensions of 130 × 6.5 × 3 mm^3^, according to ASTM D 2863-08. The UL 94 vertical burning test was carried out using a CZF-3 instrument (Jiangning Analysis Instrument Factory, Jiangning, China) with specimen dimensions of 125 × 12.5 × 3.2 mm^3^, according to GB/T 2408-2008. The cone calorimeter test (CCT) was performed with a FTT apparatus (Fire Testing Technology, London, UK) under 50 kW/m^2^ external radiant heat flux conforming to ISO 5660 protocol. The specimen dimension is 100 × 100 × 3 mm^3^. Raman spectra of residual char were measured at room temperature using a 532 nm laser (Horiba Jobin Yvon HR Evolution, Paris, France).

## 3. Results and Discussion

### 3.1. Assembly Mechanism and Structure Characterization of APP-ZIFs

As the illustration of the interface assembly mechanism of APP-ZIFs shows in Figure 1, through adjusting the immersed content of ZIF-67 and the volume ratio of water/ethanol in the solution, the ZIF-67 particles were firstly adhered onto the surface of APP particles due to the H-bond effect. After that, because the interfacial assembly was taking place in water solution, weakly acidic APP and alkalescent 2-MIZ from ZIF-67 would accelerate each other’s hydrolysis reactions. The hydrolysis of NH_4_^+^ promoted the protonation of 2-MIZ and etching of the ZIF-67, resulting in the decomposition of ZIF-67 and removal of NH_3_. The series of reaction also etched the surface of APP, making the surface rough to increase the specific surface area. Furthermore, the residual Co ions were bonded with APP by formatting a P–O–Co ionic band. Nevertheless, with the increasing immersed content of ZIF-67, the volume ratio of water/ethanol decreased, resulting in weaker hydrolysis reactions. Then, an increasing number of ZIF-67 particles, which formed complete frameworks gradually, would be adhered onto the surface of APP. The surface of assembled APP filled of ZIF-67 bulges became much rougher, exhibiting a further high specific surface area to endow the polymer with high migration resistance and flame-retardant efficiency.

#### 3.1.1. Assembly Mechanism and Structure Characterization of APP-ZIFs

Firstly, we employed XPS to detected the chemical environment of Co for indicating the interface interaction between APP and ZIF-67. As shown in Figure 2a, the Co 2p_3/2_ peak positions of APP-ZIFs ranged from 782.0 to 781.3 ev. Compared to the standard Co 2p_3/2_ peak of Co(NO_3_)_2_, Co_3_(PO_4_)_2_, and ZIF-67 (Figure 2b), the Co 2p_3/2_ peak of APPZ1 was closest to the mixture of Co(NO_3_)_2_ and Co_3_(PO_4_)_2_, while the Co 2P spectrum of APPZ4 was closest to ZIF-67. It indicated that the Co element in the APPZ1 surface did not exist as a ZIF-67 structure, while the APPZ4 surface successfully assembled ZIF-67 particles with a complete structure. As supplement evidence, we present the typical diffraction peak of ZIF-67 (12.7 °C) by XRD as shown in Figure 2c and the infrared absorption peaks of ZIF-67 (1340, 1172 and 1140 cm^−1^) by FTIR as shown in Figure 2d; these both appeared in the APPZ4 sample while not obviously in APPZ1, which was in accordance with the XPS results. 

Meanwhile, the surface relative atom concentration of typical elements P, N, and Co were calculated as listed in Table 1. The N concentration of APPZ1 obviously decreased more than APP, arising from the reaction between the ammonium group (APP) and 2-methylimidazole (ZIF-67) precursor, promoting the removal of ammonia gas. The N concentration of the other three APP-ZIFs gradually increased with the increasing immersed content of ZIF-67 precursor solution, which was caused by ratio of water/ethanol decreasing (APPZ1→APPZ4, 1:2.5→1:10) during the assembly process, limiting the ionization of APP and removal reaction of ammonia gas. When it comes to Co, the lower surface Co content of APPZ2 than APPZ1 was caused by the different environment of cobalt: Co of APPZ2 prefers to enclose into the ZIF-67 framework, while Co of APPZ1 explores the surface. Thus, the increasing content of ZIF-67, APPZ3, and APPZ4 was naturally comprised of more Co. Meanwhile, in the ICP-MS test, the bulk content of the Co element gradually increased from APPZ1 to APPZ4, as listed in Table 1. 

#### 3.1.2. Morphology and Specific Surface Area Analysis

As shown in Figure 3, ZIF-67 exhibits a cube structure (lateral size about 1 μm), while untreated APP exhibits a rod-like structure with a smooth surface (length size about 30 μm). However, rod-like APPZ1 turned to rough after the interaction between ZIF-67 and APP. With the increasing assembly content, numerous ZIF-67 nanoparticles were anchored on the surface of APPZ4. Similarly, as shown in the TEM images, the APPZ4 have many small and dark cube particles attached to the surface, directly indicating that ZIF-67 is forming on the surfaces of APP. In addition, ZIF-67 had a high specific surface area (1728.0 m^2^/g by BET), which also might exert a higher surface area when assembling onto APP. As the N_2_ adsorption/desorption isotherms for the surface area shown in Figure 4 indicate, the BET specific surface area of APPZ1, which had a surface interaction with ZIF-67, increased to 6.1 m^2^/g from 0.9 m^2^/g of untreated APP. Furthermore, the specific surface area of APPZ4, whose surface assembled an amount ZIF-67 particles with complete structure, increased to 82.9 m^2^/g, which was 92.1 times as much as that of APP. The high specific surface area of APPZ4 had great significance for promoting the charring and flame-retardant ability as well as improving the compatibility and mechanical properties when used in an EVA matrix.

### 3.2. Thermal Degradation Behaviors of APP-ZIFs with DPER as an IFR System

We employed TGA to study the thermal degradation behaviors and thermal stability of APP and APP-ZIFs when compounded with the traditional charring agent DPER in the ratio of 3:1 under nitrogen and air atmosphere. As shown in Figure 5, compared to APP/DPER, the APP-ZIFs/DPER mixture retained more char residue at high temperature both in nitrogen (>500 °C) and air (>600 °C) atmosphere. In addition, the residual mass increased with the increasing assembled ZIF-67 content, while the peak of the mass loss rate gradually decreased. It indicated that the APP-ZIFs/DPER mixture exhibited enhanced thermal stability, especially at high temperature. The difference between APP/DPER and APP-ZIFs/DPER was mainly the content of cobalt from ZIF-67. The cobalt catalysts promote more thermal-stable products at high temperature, which may also contribute to forming more char residue when used in polymers during combustion tests. Thus, we attempted to calculate the unit catalytic efficiency of cobalt, which increased the residuals at 800 °C. As listed in Table 2, under non-oxidization degradation, the unit catalytic efficiency of cobalt (Δ*R*/*M*c_o_) of APPZ4/DPER was the highest. Under oxidization degradation, APPZ1/DPER at low ZIF-67 content exerted the highest unit catalytic efficiency, while APPZ4/DPER came second. In consequence, APPZ1/DPER and APPZ4/DPER, which exerted the best unit catalytic efficiency under non-oxidization and oxidization, were chosen as flame retardants into EVA composites to investigate the improved flame retardancy endow by assembling ZIF-67 on APP. 

### 3.3. Interfacial Interaction, Mechanical Properties, and Migration Resistance

When the APP-ZIFs/DPER system has been for the comparative study, we investigated the rheological properties, mechanical properties, and migration resistance of EVA composites with 25% mass fraction flame retardants (named as 25% APP/DPER, 25% APPZ1/DPER, and 25% APPZ4/DPER). Firstly, Figure 6a displayed the complex viscosity versus the angular frequency curves (η*) of EVA composites, which responded to the interfacial interaction between flame retardants and EVA matrix in the molten state. Among all samples, 25% APPZ4/DPER, which was comprised of APPZ4 with the highest specific surface area, showed the highest complex viscosity. It happens frequently that the comparative high viscosity of EVA/APP/DPER composites brings some advantages in enhancing mechanical properties, droplet inhibition, and char forming. In Figure 6b, the enhanced interfacial interaction and compatibility between ZIF-67-treated APP and the EVA matrix were further proved by the SEM photos of brittle fracture (liquid nitrogen immersion). Different from the APP in 25% APP/DPER, which was separated, APPZ1 in 25% APPZ1/DPER and APPZ4 in 25% APPZ4/DPER were well coated by a carbonaceous EVA matrix (as can be seen in the EDS test). 

In case of the enhanced interfacial interaction and compatibility, the mechanical properties of 25% APPZ1/DPER and 25% APPZ4/DPER were much better than that of 25% APP/DPER, as shown in Figure 6c. Especially for 25% APPZ4/DPER, comparing to 25% APP/DPER, it increased the tensile strength by 27.2% (from 11.4 to 14.5 MPa), increased the tensile stress at yield by 22.4% (from 5.8 to 7.1 MPa), and increased elongation at break by 12.1% (from 572% to 641%). The significant improvement was because the high specific surface area of APPZ4 could implant into the EVA matrix and allow an increasing contact area. During tensile testing, the applied stress might be transferred from the EVA matrix onto the APPZ4 particles, resulting in a significant enhancement of tensile strength. In addition, ZIF-67 particles might separate from APPZ4 particles while suffering from the stress and initiate crazes to toughen EVA composites. However, the presence of too many crazes by ZIF-67 particles also would evolve into cracks, which was why the elongation at break of 25% APPZ4 was lower than that of 25% APPZ1/DPER. Furthermore, we employed the water resistance tests to evaluate the effect of interface assembly of ZIF-67 on inhibiting the emigration of APP from the EVA matrix. As the mass loss percentage curves shown in Figure 6d indicate, 25% APPZ4/DPER exhibited the minimum mass loss among all samples, which barely increased after 9 days. It seemed that the assembly of ZIF-67 onto APP benefited from inhibiting the emigration of APP, and this was consistent with the above discussion. In addition, as the SEM photos of surface EVA composites shown in Appendix A indicate, 25% APPZ4/DPER indeed exhibited the fewest holes by emigration of APP. Thus, the interface assembly of ZIF-67, which increases the specific surface area of APP and then the interfacial interaction of APP/EVA matrix, can enhance the water resistance of EVA composites with APP-based flame retardants. It provided another path to solve the emigration of APP against some extreme environment. 

### 3.4. Flame Retardancy and Smoke Suppression

#### 3.4.1. LOI and UL 94 Test

As shown in Table 3, the LOI value of pure EVA was only 19.0%, while those of the EVA composites with 25%, 28% and 30% APP/DPER intumescent flame retardants increased to 26.2%, 28.2%, and 29.3%, respectively. Meanwhile, 25% APP/DPER and 28%APP/DPER passed the UL 94 V-2 rating, and 30%APP/PDER passed the V-0 rating without dripping, while the neat EVA composites were unrated. The APP-ZIFs/DPER system into EVA composites can further improve the flame-retardant effect. The 25% APPZ1/DPER and 25% APPZ4/DPER samples had increased LOI values of 28.4% and 29.4%, and they passed the UL 94 V-0 rating without dripping. Surprisingly, 25% APPZ4/DPER exhibited a similar flame-retardant effect to the EVA composites with APP/DPER of 30% mass fraction.

#### 3.4.2. CCT

Cone calorimeter (CCT) was conducted to investigate the flame-retardant behaviors of EVA composites. The curves of heat release rate (HRR), total release rate (THR), smoke production rate (SPR), mass loss curves, and digital photos of residual char are shown in Figure 7, while the main characteristic parameters, such as the peak of HRR (PHRR), THR, peak of SPR (PSPR), peak CO production rate of SPR (PCOP), total smoke production (TSP), and average value of effective heat of combustion (av-EHC), are summarized in Table 4.

As shown in Figure 7 and Table 4, loading 25% APP/DPER flame retardant into EVA can obviously decrease the PHRR and THR. Furthermore, the APPZ1/DPER and APPZ4/DPER system with ZIF-67-assembled APP exerted a better flame-retardant effect, which further decreased the PHRR by 18.3% and 34.7%, decreased the THR by 5.1% and 11.4% compared with the APP/DPER system at same content, respectively. Meanwhile, 25% APPZ1/DPER and 25% APPZ4/DPER composites also decreased PSPR by 22.0% and 39.0%, and they decreased TSP by 12.4% and 22.8%, which indicated the improved smoke suppression effect. Comparing to 25% the APP/DPER composites, the residues of the 25% APPZ1/DPER and 25% APPZ4/DPER composites steeply increased by 49.2% and 91.7%, while the average value of effective heat of combustion (av-EHC) values slightly increased, indicating that the flame-retardant effect of APP-ZIFs/DPER were major in charring in the condensed phase.

In the condensed phase, the reinforced charring effect can be further proved by the digital photos of residue and mass loss curves, as shown in Figure 7. The surface char layer of 25% APPZ1/DPER and 25% APPZ4/DPER gradually became complete and compact compared to the big holes and cracks in 25% APP/DPER, indicating their ability to durably protect under the EVA matrix during combustion. Especially for 25% APPZ4/DPER, its char layer was compact without any cracks, resulting in the highest char residues. In the gas phase, the increasing av-EHC and decreasing PCOP values of 25% APPZ1/DPER and 25% APPZ4/DPER composites indicated that the decomposing volatiles from EVA underwent more complete combustion, which was beneficial to the reduction of CO and smoke toxicity. To sum up, the APPZ4/DPER system can endow EVA composites with the best flame-retardant and smoke suppression effects. 

### 3.5. Thermal Stability and Decomposing Volatiles of EVA Composites 

The thermal degradation behaviors of different EVA composites were measured by TGA under nitrogen and air atmosphere, as shown in Appendix A. At both nitrogen and air atmosphere, the 25% APPZ4/DPER retained 62.2% (N_2_) and 43.0% (Air) more residues at 700 °C than 25% APP/DPER, exerting best charring effect both under non-oxidization and oxidization degradation, which was in accordance with the above LOI, UL 94 and CCT results. In addition, we employed the quantitative TG-FTIR to analyze the composition of thermal-decomposing volatiles derived from 25% APP/DPER, 25% APPZ1/DPER, and 25% APPZ4/DPER at air atmosphere. As indicated the FTIR spectra of the volatile form of EVA composites at the maximum mass loss rate shown in Figure 8a, several typical peaks are assigned to characteristic pyrolysis products as follows: hydrocarbons (2930 cm^−1^), CO_2_ (2360 cm^−1^), CO (2190 cm^−1^), and aromatic compounds (1510 cm^−1^). In addition, the intensity of each typical pyrolysis products versus time curves is presented in Figure 8b–f. Comparing to those of 25% APP/DPER, the maximum absorbance intensity of the characteristic volatile products of 25% APPZ1/DPER and 25% APPZ4/DPER, including hydrocarbons, CO, and aromatic compounds, all significantly reduced, while only CO_2_ (complete degradation product) increased. The absorption intensity of the above flammable volatile products of 25% APPZ4/DPER was the lowest among all the EVA composites, which contributes to reducing the heat release. Furthermore, the highest CO_2_ intensity of 25% APPZ4/DPER indicated the catalytic oxidation of cobalt from ZIF-67. Meanwhile, the main toxic substance (CO) is a light-extinction precursor (aromatic compounds) in the combustion process [38], and it obviously decreased, which was beneficial in reduction of smoke toxicity. 

### 3.6. Micromorphology and Chemical Structure of the Char Residues

We employed SEM and TEM to clarify the reasons for the increasing mass and intensity of the char residue of 25% APPZ4/DPER compared with that of 25% APP/DPER. As the SEM photos show in Figure 9, the surface char layer of 25% APP/DPER exhibited an amount amorphous holes, while 25% APPZ1/DPER and 25% APPZ4/DPER became smooth with regular frameworks, indicating the better barrier effect against exterior heat and oxygen. The assembled frameworks were in circle or hexahedron shapes, and the cross-linked structures were rich in Co, as detected by EDS (Appendix A). As shown in the low-magnification TEM photos, the surface char of 25% APP/DPER was made up of a carbonaceous structure and irregular floccule. However, the char of 25% APPZ4/DPER was comprised of a large amount of nano spherical frameworks and a graphene-shaped 2D nano film structure. The 2D nano films were identified as cobalt phosphate by EDS (Appendix A). These nano cobalt phosphates had a high surface area and catalytic efficiency, leading to the higher catalytic graphitization of 25% APPZ4/DPER compared with that of 25% APP/DPER. 

As the high-magnification TEM images show in Figure 10a,b, a greater amount of increasing graphite-like carbon can be observed in the 25% APPZ4/DPER than in the 25% APP/DPER. As evidence, the intensity ratio of A_D_/A_G_ by Raman analysis, which was associated with the graphite carbon and the unorganized carbon structure respectively, could be used to characterize the defects in the graphite planes or the extent of graphitization in the 25% APP/DPER and 25% APPZ4/DPER composites. As shown in Figure 11a, comparing to 25% APP/DPER, the A_D_/A_G_ of 25% APPZ4/DPER decreased from 1.60 to 1.16, indicating the higher graphitization degree of 25% APPZ4/DPER. The assemble cobalt phosphate (Figure 10c,d) was the key that promoted the high graphitization and the strong protective char barrier in Figure 7. With the obviously enhanced char barrier, the 25% APPZ4/DPER retained the most char residue and exerted the best flame-retardant effect. In addition, after dealing with the Co 2p_3/2_ curve by XPS with peak differentiating and imitating, we found that various chemical environments of Co can be observed in the surface char of 25% APPZ4/DPER composites (Figure 10b). The Co element was of a different valence state, including cobalt, cobalt oxides (CoO and Co_2_O_3_), and cobalt salts (Co_3_(PO_4_)_2_ and Co(NO_3_)_2_) (the same as shown in Figure 2b), indicating the reductive coupling reaction between the cobalt compound and carbon compound. During the catalytic graphitization of carbon from EVA and DPER, the cobalt in APPZ4 was deoxidized into a lower valence state. However, during combustion under air atmosphere, the low-valence state cobalt was easily oxidized into bivalent and trivalent cobalt again. As the redox cycle proceeded, increasing thermal-stable graphite carbon was forming in the surface barrier. Furthermore, due to the char barrier with cobalt phosphate catalysts, the pyrolyzed fragments underwent a more thermal oxidized interaction, resulting in more complete combustion and less light-extinction precursors for smoke and toxic gas. 

### 3.7. Flame-Retardant and Smoke Suppression Mechanism 

According to all the discussed results, the flame-retardant and smoke suppression mechanism of EVA composites with well-dispersed and high specific surface area APPZ4/DPER can be concluded in Figure 12. During combustion, the APPZ4/DPER system gradually decomposed and promoted the formation of an intumescent char layer. With the increasing temperature (above 500 °C), the cobalt redoxed with the pyrolytic fragments and catalyzed these fragments into a graphite-like char layer. During the cycle redox, the APPZ4 transformed into two nano cobalt phosphate catalysts: 2D nano film and aggregate nano frameworks, which can exert high catalytic graphitization efficiency. The generating surface char barrier was comprised of a large amount of graphite-like carbon, which can durably protect against fire under the EVA matrix. The barrier promoted a more thermal-oxidized interaction of the pyrolyzed carbonaceous groups, resulting in more complete combustion and less light-extinction precursors for smoke and toxic CO. To sum up, the APPZ4/DPER system can endow EVA composites with an enhanced effect on both flame retardancy and smoke suppression. 

## 4. Conclusions

In summary, we proposed a feasible strategy for prepared APP with a high specific surface area (APP-ZIFs) by assembling ZIF-67, which opens a new avenue for preparing APP with both high flame-retardant efficiency and migration resistance. The assembly mechanism, morphology, chemical structure, and compositions of APP-ZIFs were characterized. Comparing to EVA composites with 25% APP/DPER, 25% APPZ4/DPER effectively improved the mechanical, migration-resistant, flame-retardant, smoke-suppression, and toxicity-suppression properties. These improvements were attributed to the catalytic graphitization by nano cobalt phosphate catalysts derived from APPZ4 and the more protective char barrier of graphite-like carbon. 

## Figures and Tables

**Figure 1 polymers-12-00534-f001:**
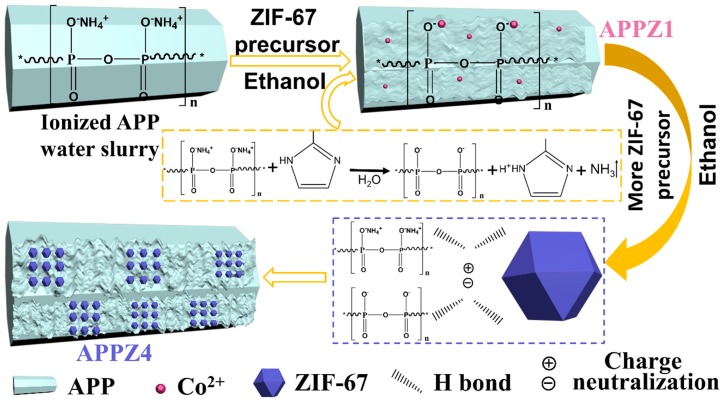
Illustration of the interface assembly mechanism of ammonium polyphosphate zeolite imidazole frameworks (APP-ZIFs).

**Figure 2 polymers-12-00534-f002:**
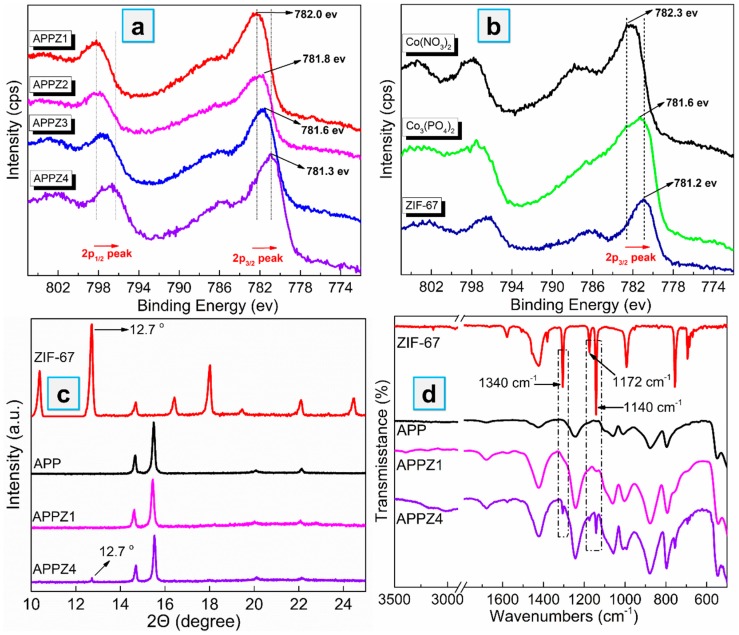
Co 2P X-ray photoelectron spectroscopy (XPS) spectrum (**a**,**b**), XRD patterns (**c**), and Fourier transform infrared (FTIR) spectra (**d**) of ZIF-67 and APP-ZIFs.

**Figure 3 polymers-12-00534-f003:**
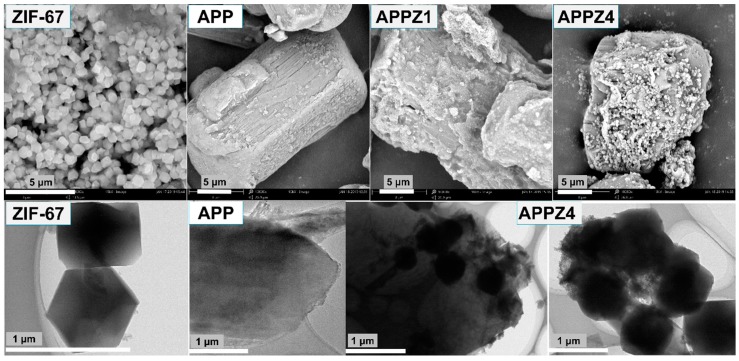
SEM images of the ZIF-67, APP, APPZ1, and APPZ4 (first line); TEM images of ZIF-67, APP, and APPZ4 (second line).

**Figure 4 polymers-12-00534-f004:**
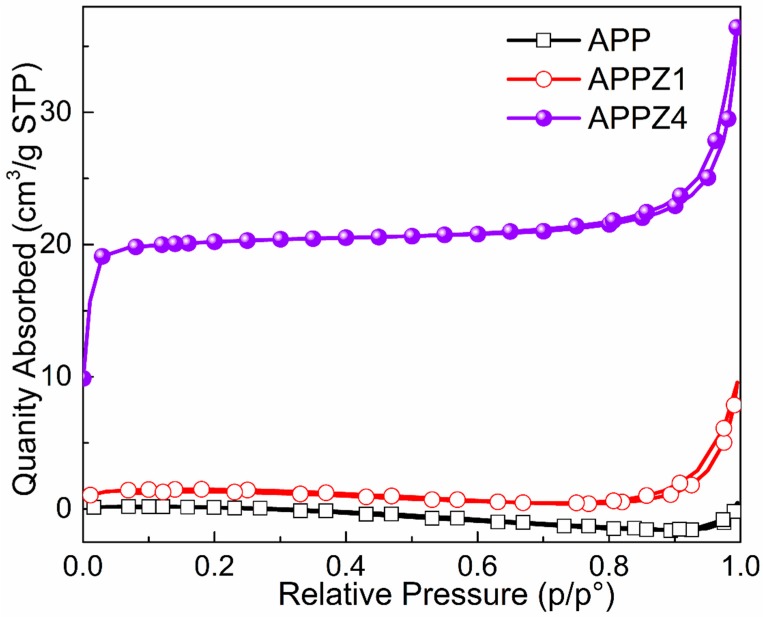
N_2_-sorption isotherms for APP-ZIF particles.

**Figure 5 polymers-12-00534-f005:**
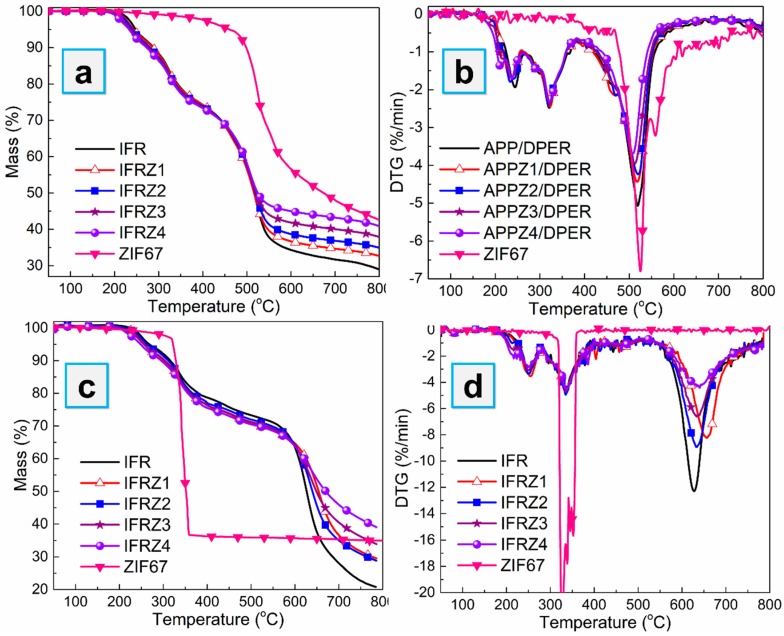
Thermogravimetric (TG) (**a**) and DTG (**b**) curves of APP-ZIFs/DPER (dipentaerythritol) in N_2_; TG (**c**) and DTG (**d**) curves of APP-ZIFs/DPER in air.

**Figure 6 polymers-12-00534-f006:**
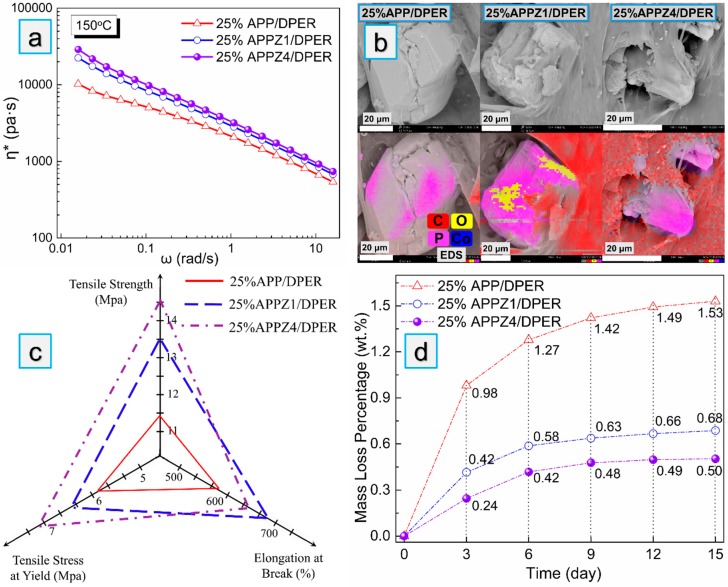
Complex viscosity (η*) versus the angular frequency curves (**a**), SEM photos, and energy-dispersive spectrometer (EDS) analysis (**b**), mechanical properties (**c**) and water- resistance tests (**d**) of ethylene-vinyl acetate (EVA) composites.

**Figure 7 polymers-12-00534-f007:**
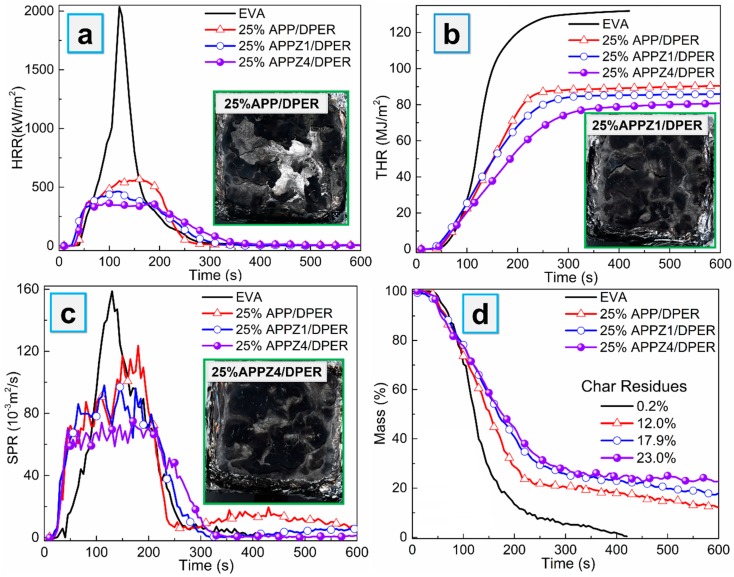
The heat release rate (HRR) (**a**), total release rate (THR) (**b**), smoke production rate (SPR) (**c**), mass loss curves (**d**), and the inset pictures in (**a**–**c**) were the digital photos of residual char of EVA composites from cone calorimeter.

**Figure 8 polymers-12-00534-f008:**
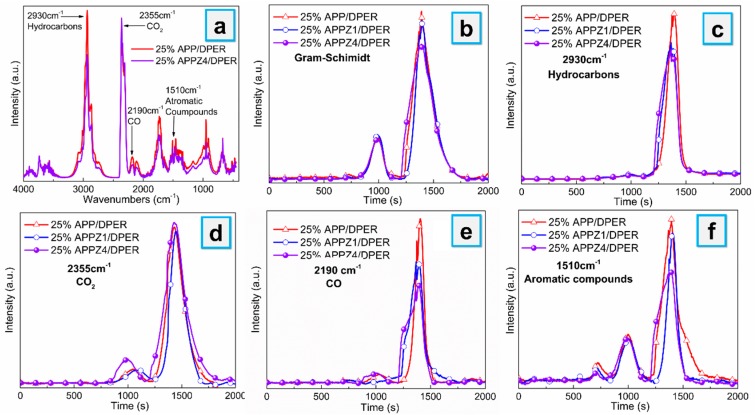
FTIR spectra of volatility at the maximum mass loss rate (**a**); absorbance of volatile products of EVA composites vs. time: (**b**) total pyrolysis products, (**c**) hydrocarbons, (**d**) CO_2_, (**e**) CO, and (**f**) aromatic compounds.

**Figure 9 polymers-12-00534-f009:**
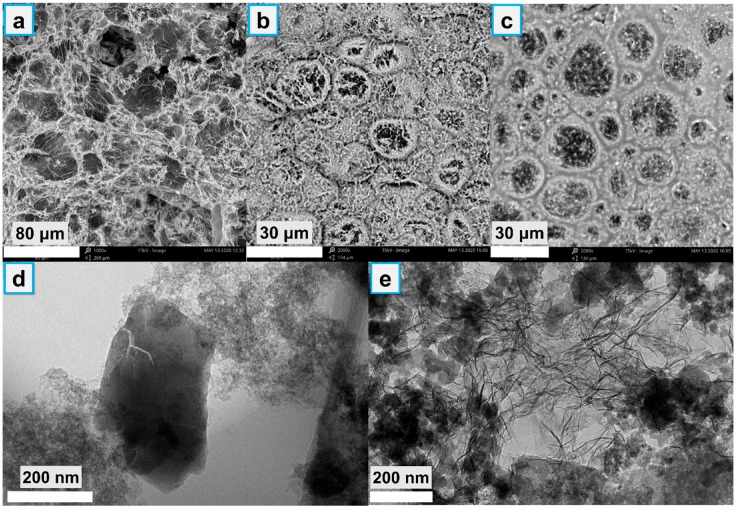
SEM photos of surface char layer of 25% APP/DPER (**a**), 25% APPZ1/DPER (**b**), 25% APPZ4/DPER (**c**); low-magnification TEM images of a surface char layer of 25% APP/DPER (**d**), and 25% APPZ4/DPER (**e**) composites.

**Figure 10 polymers-12-00534-f010:**
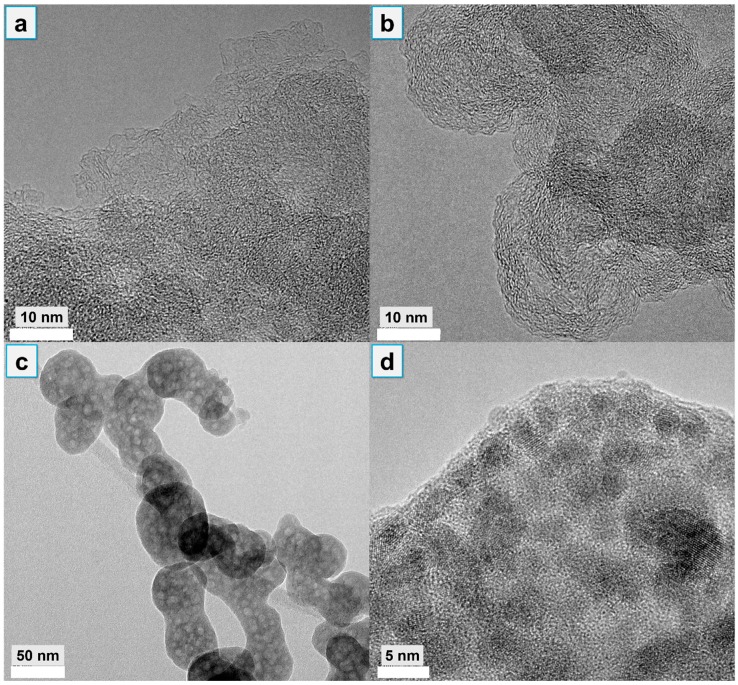
High-magnification TEM images of surface char: C-rich region in 25% APP/DPER (**a**) and 25% APPZ4/DPER (**b**); Co-rich region in 25% APPZ4/DPER (**c**,**d**).

**Figure 11 polymers-12-00534-f011:**
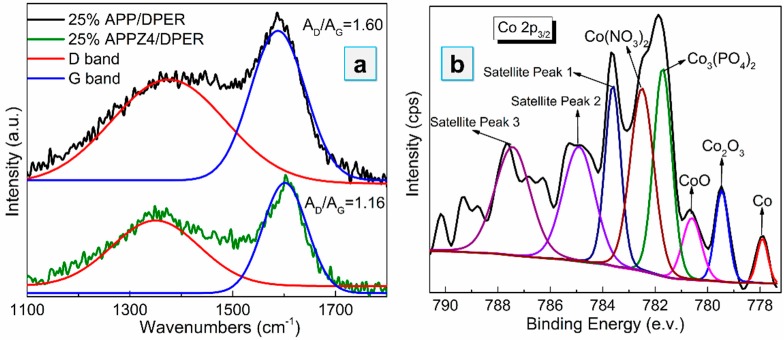
Surface char analysis: (**a**) Raman analysis of 25% APP/DPER and 25% APPZ4/DPER; (**b**) fitting curves of XPS Co 2p_3/2_ spectrum of 25% APPZ4/DPER.

**Figure 12 polymers-12-00534-f012:**
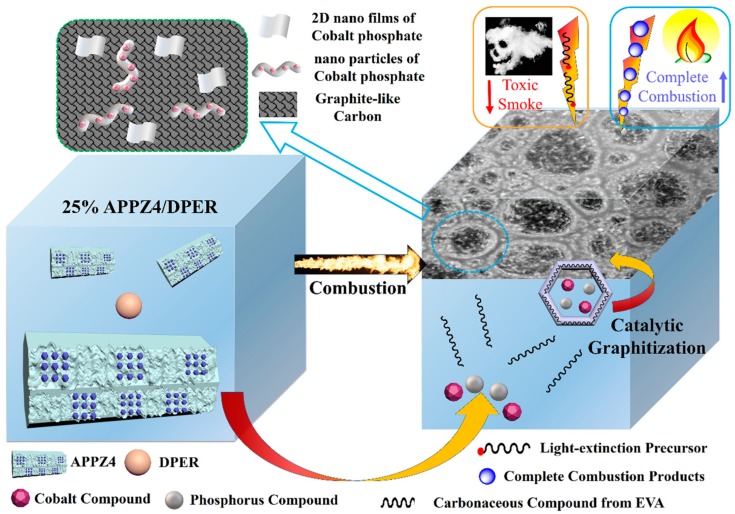
Illustration of the flame-retardant and smoke suppression mechanism.

**Table 1 polymers-12-00534-t001:** Surface relative atom concentration of element and bulk element content.

Samples	XPS (ac.%)			ICP-MS (wt %)
	P	N	Co	Co
APP	21.8	78.2	0.0	--
APPZ1	31.6	43.4	25.0	0.76
APPZ2	31.2	47.6	21.2	1.15
APPZ3	20.0	50.3	29.7	1.52
APPZ4	13.8	52.6	33.6	1.82

**Table 2 polymers-12-00534-t002:** Thermal stability of APP-ZIFs/DPER at 800 °C.

Samples		N_2_ Atmosphere			Air Atmosphere	
	^a^*R* 800°C (%)	^b^ Δ*R* (%)	^c^ Δ*R*/*M*c_o_ (%/%)	*R* 800°C (%)	Δ*R* (%)	Δ*R*/*M*c_o_ (%/%)
APP/DPER	29.0	--	--	20.8	--	--
APPZ1/DPER	32.7	3.7	4.9	29.5	8.7	11.4
APPZ2/DPER	35.0	6.0	5.2	28.8	8.0	7.0
APPZ3/DPER	38.0	9.0	5.9	33.9	13.1	8.6
APPZ4/DPER	41.0	12.0	6.6	39.0	18.2	10.0
ZIF-67	42.7		--	35.0		--

^a^*R*-800 °C: residuals at 800 °C; ^b^ Δ*R*: residuals of APP-ZIFs/DPER minus residuals of APP/DPERR; ^c^Δ*R*/*M*c_o_: Δ*R* of unit mass Co (1 wt %), Δ*R*/*M*c_o_ = Δ*R*/*M*_(cobalt)._

**Table 3 polymers-12-00534-t003:** Limiting oxygen index (LOI) and UL 94 results of EVA composites.

Samples	LOI (%)	UL 94 Rating (3.2 mm)	*t*_1_/*t*_2_^#^ (s)	Dripping
EVA	19.0	NR	-- *	Y
25% APP/DPER	26.2	V-2	0.8/12.4	Y
28% APP/DPER	28.2	V-2	0.9/4.1	Y
30% APP/DPER	29.3	V-0	0.7/2.1	N
25% APPZ1/DPER	28.4	V-0	0.7/4.7	N
25% APPZ4/DPER	29.4	V-0	0.7/4.3	N

^#^*t*_1_/*t*_2_ were the average burning time after 10 s flame; * The samples were burning seriously with fire dripping until the end.

**Table 4 polymers-12-00534-t004:** The data from CCT of EVA composites. PHRR: peak of HRR, PSPR: peak SPR, TSP: total smoke production, PCOP: peak CO production rate of SPR.

Samples	PHRR (kw/m^2^)	THR (MJ/m^2^)	PSPR (10^−3^ m^2^/s)	TSP (m^2^)	av-EHC (MJ/kg)	PCOP (10^−3^ m^2^/s)
EVA	2037 ± 35	145.9 ± 6.2	159 ± 7	18.5 ± 0.9	48.0 ± 1.3	17.6 ± 0.9
25% APP/DPER	568 ± 22	90.5 ± 2.1	123 ± 4	20.2 ± 1.0	30.9 ± 0.5	14.2 ± 0.7
25% APPZ1/DPER	464 ± 20	85.9 ± 2.2	96 ± 3	17.7 ± 0.6	31.5 ± 0.4	12.2 ± 0.6
25% APPZ4/DPER	371 ± 18	80.2 ± 1.9	75 ± 3	15.6 ± 0.5	33.3 ± 0.5	8.7 ± 0.5

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
