# Peer review of "Ammonium Polyphosphate with High Specific Surface Area by Assembling Zeolite Imidazole Framework in EVA Resin: Significant Mechanical Properties, Migration Resistance, and Flame Retardancy"

_polymers, 2020, doi:10.3390/polym12030534_

Round 1

Reviewer 1 Report

The paper is original, globally clear with a good scientific level. It could be accepted after some minor correction. Please check the enclosed file containing some comments and correction suggestions.

Author Response

Thanks for your comments very much. Your comments are of great significance to us that pointed out the deficiencies of our manuscript. After understanding your precious comments carefully, we have revised the manuscript according to your enclosed file. Please see the attachment.

Reviewer 2 Report

Obviously, this work is well designed and processed. It mainly introduces the design of ZIF combined with a traditional APP that could achieve satisfying enhancement of EVA. The characterizations can fully verify the properties of the obtained samples. The results including properties and mechanism investigation are well done and support the view. I recommend that this paper could be published in Polymers after major revision. The suggestions are listed below:

For technical issues,

  1. First, a paper with well-written is important for readers. This paper has a lot of grammar mistakes hardly meeting the requirement of Polymers. This paper should be revised by a native English speaker.
  2. The figures including the curves and words shown in the manuscript are rough and hard to recognize, authors should revise them and provide high-quality figures. This is very important.
  3. Moreover, in the case of the TEM and/or SEM images, the color of the scale bar and the numbers should be identical and have better using white or black. This will be more visible.
  4. It is worth noting that in a manuscript, it is not suitable to use a, b, c and meanwhile a, a’, b, c. This kind of written style is not standard and can easily make readers confused. Please check and revise them.
  5. In Figure 6, the green frame is not suitable, I suggest authors remove it. Additionally, in Figure 6b, you should give the introduction of the EDS images in the figure caption, including the color meaning. The scale bar is too small to identify.
  6. For tables, for instance, table 3, the line of “25% APP/DPER”, authors should adjust the “0.8/12.4”, making the line space identical. Please check all the tables.
  7. For Figure 8, this figure is too small to recognize the content. The vertical axis “intensity (a.u.) could be removed.
  8. Please remove the green frame of Figure 10, and make Figure 10c, f more visible. Please check all the figures.

For research content,

  1. In 3.2 section, it is not scientific to use TGA to investigate the charring ability of polymers. As we know, TGA is a technology to analyze thermal stability. The samples are degraded to pieces under increasing temperature no matter under nitrogen or air environment. The samples are not transferred to char, moreover, the weight of the testing sample is too little, it cannot reflect the charring ability of polymers. Therefore, authors should completely revise this section, provide a reasonable explanation.
  2. Reasonable explanations for the TGA results are necessary including comparison with reported works.
  3. The case of phosphorus contained flame retardancy, some reported works are worthy cited, such as “Industrial & Engineering Chemistry Research 56 (23), 6664-6670” and “Composites Part A: Applied Science and Manufacturing 94, 170-177”.
  4. In Figure 8, 9, 10, the TG-IR, SEM of char residue, and Raman data for pristine EVA should be provided and analysed.

Author Response

Thanks for your comments very much. Your comments are of great significance to us that pointed out the deficiencies of our manuscript. After understanding your precious comments carefully, we have revised the manuscript. We have revised the manuscript considering your previous comments. And all the revised content in the revised manuscript were marked in red color. Thanks for your appreciate time.

Reviewer 3 Report

Very good paper, well done. Can be published, I would only recommend language checking.

Author Response

Thanks for your comments very much.  We have revised the manuscript considering your previous comments. 

Round 2

Reviewer 2 Report

The authors have revised the questions I raised. To my view, this paper could be published.